# Effect of Lipid-Based Multiple Micronutrients Supplementation in Underweight Primigravida Pre-Eclamptic Women on Maternal and Pregnancy Outcomes: Randomized Clinical Trial

**DOI:** 10.3390/medicina58121772

**Published:** 2022-11-30

**Authors:** Nabila Sher, Murad A. Mubaraki, Hafsa Zafar, Rubina Nazli, Mashal Zafar, Sadia Fatima, Fozia Fozia

**Affiliations:** 1Biochemistry Department, Khyber Girls Medical College, Peshawar 25000, Pakistan; 2Department of Biochemistry, Institute of Basic Medical Science, Khyber Medical University, Peshawar 25000, Pakistan; 3Clinical Laboratory Sciences Department, College of Applied Medical Sciences, King Saud University, Riyadh 11433, Saudi Arabia; 4Department of Biochemistry KMU Institute of Medical Science, Khyber Medical University, Kohat 26000, Pakistan

**Keywords:** pre-eclampsia, lipid-based nutritional supplements, pregnancy outcome, maternal outcome, Khyber Pakhtunkhwa Province of Pakistan

## Abstract

*Background and Objectives:* In pre-eclampsia, restricted blood supply due to the lack of trophoblastic cell invasion and spiral artery remodeling is responsible for adverse pregnancies and maternal outcomes, which is added to by maternal undernutrition. This study was designed to observe the effect of multiple nutritional micronutrient supplements on the pregnancy outcomes of underweight pre-eclamptic women. To investigate the effects of lipid-based multiple micr supplementations (LNS-PLW) on pregnancy and maternal outcomes in underweight primigravida pre-eclamptic women. *Materials and Methods:* A total of 60 pre-eclamptic, underweight primigravida women from the antenatal units of tertiary care hospitals in the Khyber Pakhtunkhwa Province, Pakistan, were randomly divided into two groups (Group 1 and Group 2). The participants of both groups were receiving routine treatment for pre-eclampsia: iron (60 mgs) and folic acid (400 ug) IFA daily. Group 2 was given an additional sachet of 75 gm LNS-PLW daily till delivery. The pregnancy outcomes of both groups were recorded. The clinical parameters, hemoglobin, platelet count, and proteinuria were measured at recruitment. *Results:* The percentage of live births in Group 2 was 93% compared to 92% in Group 1. There were more normal vaginal deliveries (NVDs) in Group 2 compared to Group 1 (Group 2, 78% NVD; group 1, 69% NVD). In Group 1, 4% of the participants developed eclampsia. The frequency of cesarean sections was 8/26 (31%) in Group 1 and 6/28 (22%) in Group 2. The number of intrauterine deaths (IUDs) was only 1/28 (4%) in Group 2, while it was 2/26 (8%) in Group 1. The gestational age at delivery significantly improved with LNS-PLW supplementation (Group 2, 38.64 ± 0.78 weeks; Group 1, 36.88 ± 1.55 weeks, *p*-value 0.006). The Apgar score (Group 2, 9.3; Group 1, 8.4) and the birth weight of the babies improved with maternal supplementation with LNS-PLW (Group 2, 38.64 ± 0.78 weeks: Group 1, 36.88 ± 1.55; *p*-value 0.003). There was no significant difference in systolic blood pressure, while diastolic blood pressure (Group 2, 89.57 ± 2.08 mmHg; Group 1, 92.17 ± 5.18 mmHg, *p*-value 0.025) showed significant improvement with LNS-PLW supplementation. The hemoglobin concentration increased with the LNS-PLW supplement consumed in Group 2 (Group 2, 12.15 ± 0.78 g/dL; Group 1, 11.39 ± 0.48 g/dL, *p*-value < 0.001). However, no significant difference among the platelet counts of the two groups was observed. *Conclusions:* The pregnancy and maternal outcomes of underweight pre-eclamptic women can be improved by the prenatal daily supplementation of LNS-PLW during pregnancy, along with IFA and regular antenatal care and follow-up.

## 1. Introduction

Pre-eclampsia is the most common hypertensive disorder in pregnancy. About 3–5% of first pregnancies are complicated by pre-eclampsia globally, with a near 15% recurrence rate in subsequent pregnancies [1]. It is characterized by hypertension, proteinuria, and edema ≥20 weeks of gestation in women previously normotensive [2,3]. Pre-eclampsia is a multisystem disorder with a placental origin and creates immediate and long-term fetal and maternal complications. It is documented as one cause of maternal mortality, morbidity, and stillbirths [4,5,6]. In Pakistan, approximately 19% of maternal mortality is attributed to pre-eclampsia [7], rating only third after postpartum hemorrhage and sepsis [4]. Pre-eclampsia may be subclassified as mild and severe, depending upon the presence of hypertension [1]: mild pre-eclampsia with BP (140–159/90–109 mmHg) and severe pre-eclampsia with BP (≥160/110 mmHg) [8]. On the bases of gestational age, pre-eclampsia can also be subclassified as being early-onset (≤34 weeks) due to fetal disorders, primarily through imperfect placentation. Late-onset (≥34 weeks) is due to maternal disorders, lowered maternal thresholds, and excessive physiological placentation [4]. Pre-eclampsia occurs due to abnormal trophoblastic invasion, causing reduced placental perfusion and the replacement of maternal arteriole endothelial cells through an imbalance between the angiogenic and antiangiogenic factors. This, in turn, leads to the damage and dysfunction of the endothelium of maternal arteries, leading to hypertension (due to vasoconstriction), edema due to increased vascular permeability, and proteinuria (due to glomerular damage) [1]. The angiogenic factors are essential for the vessel’s health and for new vessel formation and organ development. In pre-eclampsia, the high levels of soluble fms-like tyrosine kinase-1(sFIt-1), soluble Endoglin (sEng), and antiangiogenic factors affect the levels of placental growth factor (PIGF), transforming growth factor β (TGEβ) and vascular endothelial growth factor, respectively [1]. These low levels lead to endothelial dysfunction and the clinical characteristics of pre-eclampsia [6]. The placental growth factor (PIGF) and soluble fms-like tyrosine kinase-1(sFIt-1) are used to differentiate pre-eclampsia from other diseases, like chronic hypertension and kidney diseases [5]. The pathophysiology of pre-eclampsia occurs due to abnormal trophoblastic invasions, causing reduced placental perfusion and the replacement of maternal arteriole endothelial cells through an imbalance between the angiogenic and antiangiogenic factors [5].

Pre-eclampsia is (7.5%) more common among nulliparous women than parous women [7]. A young age, being underweight, and multiple pregnancies are also non-specific risk factors of pre-eclampsia [9]. Moreover, malnutrition, undernutrition, high-fat/fiber diets, and micronutrient deficiency are risk factors for pre-eclampsia [10,11,12]. The nutritional status of pregnant women plays a vital role in fetal growth and development [13,14]. In low-income countries, maternal undernutrition and anemia are the common causes of maternal mortality and morbidity. Women during pregnancy are at risk of nutritional deficiencies because of the added burden of the metabolism of the fetus and the growing placenta [1]. The deficiency of macro- and micronutrients is common among women of child-bearing age in developing countries [8]. The association between multiple micronutrient deficiencies and the development of pre-eclampsia has also been reported [14,15]. Therefore, undernutrition during pregnancy affects not only the mother’s health but also the health of the growing fetus. In order to balance and improve the micronutrient deficiencies and pregnancy outcomes, iron, folic acid, calcium, and multiple micronutrient supplements are used during pregnancy and before conception [4,5]. In order to improve maternal nutrition and pregnancy outcomes, lipid-based multiple micronutrient supplements are given to underweight pregnant and lactating mothers (LNS-PLW) in underdeveloped countries [6,7]. The benefits of lipid-based multiple micronutrient supplementation have been assessed in normotensive undernourished women [6]. The LNS-PLW is a peanut-based product from the World Food Program (WFP) that is packed in a ready-to-use sachet of 75 gm, having an energy of 400 kcal (Table 1). It is provided to childbearing-aged women to improve maternal nutrition and pregnancy outcomes in underdeveloped countries like Pakistan [1,8].

The World Health Organization has recommended that pre-eclampsia screening should be a part of routine antenatal investigation. Most of the routine antenatal investigations are done for the clinical condition which has already been present like screening for viral disease or ultrasound-based screening for morphological abnormalities of the fetus. There is no predictive screening for diseases that have not yet occurred or developed, for example, pre-eclampsia and eclampsia. The outcomes of such a condition are alarming once it develops; therefore, the early detection and prevention of prophylactic therapy may be beneficial. This would enable the clinician to start timely preventive measures, antenatal care, and prophylactic treatment [16,17]. Pre-eclampsia screening has no real cost or side effect from its implementation; therefore, it must be part of the routine antenatal investigation or examination [18]. There are different screening strategies for pre-eclampsia, ranging from simple maternal history analysis to early screening biomarkers.

The screening approaches, which are in practice nowadays, include a detailed maternal history [18], the measurement of mean arterial blood pressure of the pregnant mother, performing Doppler ultrasounds on uterine arteries [19], the identification of serum cardio-metabolic biomarkers [20], the identification of novel markers, like cell-free DNA [21], multiple approaches, etc. [22]. The management of pre-eclampsia is divided into three stages: prevention, early detection, and treatment. Dietary counseling, stopping smoking, the management of pre-existing hepatic and renal diseases, giving low-dose aspirin before 12 weeks of pregnancy, calcium supplementation, antihypertensive drugs, and magnesium sulfate are all prescribed for the prevention of eclampsia. Moreover, during hospitalization due to severe pre-eclampsia (for the termination of the pregnancy), a cesarean section may help in the management of the condition [23].

Pre-eclampsia is more common in primigravida, and it is associated with micronutrient deficiencies. Our country is facing a lack of a proper diet, especially for women of childbearing age, due to poverty. Nutritional deficiency is one of the risk factors of pre-eclampsia. To our knowledge, little evidence is available on the effect of LNS-PLW in pre-eclamptic pregnancy and maternal outcomes in our region. The prevalence of undernutrition at childbearing age is quite high; therefore, the current study was designed and conducted to assess the impact of lipid-based multiple micronutrient supplementation (LNS-PLW) on the maternal and pregnancy outcomes of pre-eclamptic underweight women during their first pregnancy.

## 2. Methods

After screening, 463 pre-eclamptic women from April 2018-December 2019 (Figure 1) we recruited, considering primigravida pre-eclamptic women whose BMI was less than the requirement for their gestational age retrospectively on their first antenatal visit from their antenatal record file. The BMI is calculated by using a computer pregnancy weight gain calculator. This study was conducted in the tertiary health care facilities of Peshawar (Lady Reading Hospital, Hayatabad Medical Complex) and Swat (Civil Hospital Matta), Khyber Pakhtunkhwa (KP) province of Pakistan. The trial was approved by Khyber Medical University, by the Advanced Study and Research Board, and the Ethical Committee (No: DIR/KMU-AS&RB/EN/000527-18 August 2016) (DIR/KMU = EB/EN/000314 on 27 October 2016). The participants were divided by using a computer randomizer (version 3.0) (Medical University of Graz, Institute for Medical Informatics, Statistics and Documentation Auenbruggerplatz 2/5, A-8036 Graz, Austria) into two groups (Group 1 and Group 2) and informed consent was obtained from them. The participants of both groups received conventional treatment for pre-eclampsia and IFA daily. The Group 2 participants received an additional 75 gm sachet of high-energy LNS-PLW supplement daily till delivery. One participant of Group 2 disliked the taste of LNS-PLW, and another participant lost contact and follow-up, so they dropped out. Two participants in Group 1 refused to continue, and two participants lost contact and dropped out. Therefore, 26 participants from Group 1 and 28 participants from Group 2 completed the follow-up, and we collected their pregnancy and maternal outcomes.

### 2.1. Data Collection

The socioeconomic data, family, and personal histories were recorded by predesigned questionnaires given to each participant. The age, gestational age, anthropometric measurements (height, weight, BMI), systolic and diastolic blood pressure, and the proteinuria of the participants were measured. About 5 mL of blood was taken from each participant at the time of enrollment and then delivered via an aseptic technique. The hemoglobin and platelet counts were measured by Sysmex XE-2100 hematology analyzer (Sysmex Corporation, Shisumekkusu Kabushiki-gaisha, Tokyo, Japan). The LNS-PLW was provided weekly and the empty sachets were collected to check their compliance. The leftovers in the sachet were measured. The principal researcher reminded each participant, telephonically, on alternate days to take their medication and LNS-PLW supplement. Their weight was measured by a Beurer digital glass weight scale-GS 200 Allium (Lessingstraße 10 b 89231 Neu-Ulm, Germany), and for their height, a portable stadiometer (Seca Leicester 214) (Hammer Steindamm 9-25. 22089 Hamburg, Germany) was used. The BMI kg/m^2^ of the participants was measured. At 10 min intervals, the blood pressure of the patients was measured three times by using a standard mercury sphygmomanometer, and the mean was noted. Proteinuria was measured via the dipstick method. The mode of delivery was also recorded.

### 2.2. Statistical Analysis

The Shapiro–Wilk test was used for normalizing the data; it was found to be normally distributed. The demographic data were analyzed as descriptive statistics, and the values were presented as mean ± standard deviation. The data of the two groups were compared by student t-testing. For the statistical analyses, we used SPSS software 22 (IBM, Armonk, NY, USA), and *p* < 0.05 was considered significant.

## 3. Results

The mean age of the participants (Group 1: 22.1 ± 3.3 years; Group 2: 23.3 ± 3.6 years, *p*-value 0.17) ranged from 15–35 years. Their BMI at the first antenatal visit was calculated retrospectively from their antenatal record (Group 1: 19.6 ± 1.3; Group 2: 19.7 ± 0.9, *p*-value 0.84). While the BMI kg/m^2^ at enrollment was recorded as Group 1: 22.6 ± 1.5 kg/m^2^; Group 2, 22.5 ± 1.3 kg/m^2^, *p*-value 0.459. The gestational age (from the first antenatal visit) was calculated retrospectively from participants’ antenatal records (Group 1: 19.8 ± 3.5 weeks; Group 2: 20.5 ± 3.1 weeks, *p*-value 0.45), while the gestational age at enrollment was 30.1 ± 2 weeks for Group 1 and 29.8 ± 2.3 weeks for Group 2 (*p*-value 0.59). The baseline systolic blood pressure was 146.7 ± 8.9 mmHg for Group 1 and 144.7 ± 6.8 mmHg for Group 2 (*p*-value 0.63), and the diastolic blood pressure was 95.7 ± 5.2 mmHg for Group 1 and 95.5 ± 4.4 mmHg for Group 2 (*p*-value 0.89). The proteinuria recorded via the dipstick method at enrollment was as follows: Group 1: 1.8 ± 0.7; Group 2: 1.9 ± 0.7 (*p*-value 0.34). The baseline hemoglobin concentration was as follows: Group 1: 10.5 ± 0.8 g/dL; Group 2: 10.5 ± 0.9 g/dL (*p*-value 0.92). The platelets count was as follows: Group 1: 202,166.67 ± 57,770.70/uL; Group 2: 196,200.00 ± 53,832.05/uL (*p*-value 0.68), taken during recruitment, as shown in Table 2.

The socioeconomic data of the study participants were collected; it was noted that 49% of participants belonged to the low-income group, 28% to the middle-income group, and only 23% belonged to the high-income group. The literacy rate among the participants was also found to be low: 76.66% of women in the control group and 83.33 % in the LNS-PLW group were found to be illiterate.

### Maternal Follow-Up at Delivery

The supplements consumed by the two groups during the prenatal periods (IFA by Group 1: 6.8 ± 1.9 weeks; IFA and LNS-PLW by Group 2: 8.8 ± 1.9 weeks; *p*-value 0.00).

The live birth in Group 2 was 93%, while in Group 1, it was 92%. The number of normal vaginal deliveries in Group 2 was higher: Group 1: 69% NVD; Group 2:78% NVD. In Group 1, 4% of the participants developed eclampsia as shown in Table 3.

The gestational age at delivery was higher in Group 2 (Group 1: 36.9 ± 1.6 weeks; Group 2: 38.6 ± 0.8 weeks, *p* = 0.006).

The blood pressure was calculated after delivery; the systolic BP of both groups was the same (Group 1: 135.7 ± 11.2 mmHg; Group 2: 131.3 ± 4.6 mmHg, *p*-value 0.07), while the diastolic BP (Group 1: 92.2 ± 5.2 mmHg; Group 2: 89.6 ± 2.1 mmHg, *p*-value 0.025) showed significant improvement in the LNS-PLW-consuming Group 2 as shown in Table 4.

The blood samples were collected from the women after delivery and during their stay in the hospital in order to compare the hemoglobin concentration. Hemoglobin concentration significantly improved with LNS-PLW supplementation (Group 1: 11.4 ± 0.5 g/dL; Group 2: 12.2 ± 0.8 g/dL, *p*-value 0.001).

## 4. Discussion

Pre-eclampsia is a common pregnancy-related complication that occurs in childbearing-age women. It is responsible for poor maternal and fetal outcomes, fetal and maternal mortality, and morbidity [15]. A balance nutritional intake during pregnancy is responsible for the proper functioning of bodily systems [24,25]. Pregnancy is a condition for which appropriate and balanced food intake is needed not only for the mother but also for the growing fetus. Maternal undernutrition and malnutrition increase the risk of pre-eclampsia, gestational hypertension, anemia, premature delivery, fetal and maternal mortality, and morbidity [26,27].

In this study, the hemoglobin concentration significantly increased in Group 2 when compared with Group 1. This may be due to the 10 mg of extra iron in the LNS-PLW supplement with routine IFA (60 mg iron and 400 ug folic acid). In contradiction with findings, a study was conducted in Malawi, in which three groups of women were supplemented from ≤20 weeks of gestation with IFA (containing 60 mg of iron), MMN (containing 20 mg of iron), and LNS (containing 20 mg of iron); at 36 weeks of gestation, the researchers compared the women’s hemoglobin levels and reported higher concentrations of hemoglobin in the IFA group [28]. Another study conducted in Bangladesh compared pregnant women (≤20 weeks of gestation) who were receiving either IFA (60 mg of iron and 400 ug of folic acid) or LNS (containing 20 mg of iron). They reported a higher concentration of hemoglobin in the IFA group [29]. Similarly, a study conducted on Ghanaian women reported higher hemoglobin concentrations in women using IFA (containing 60 mg of iron) when compared to those given LNS (containing 20 mg of iron) from ≤20 weeks of gestation [30]. The difference in our study was that, along with IFA (60 mg of iron), we provided an extra 10 mg of iron in the LNS-PLW as well, whereas, in the above studies, the researchers compared three groups of women with different supplements, i.e., IFA, MMN, or LNS individually.

The diastolic blood pressure of the LNS-PLW consumers in the current study decreased significantly at delivery, while no difference was observed for their systolic blood pressure. The LNS-PLW used in the current study was peanut-based, and 80% of its fat content is composed of omega-3 (60%) and omega-6 (20%). Several meta-analyses and trials reported the beneficial effects of ω-3 polyunsaturated fatty acid supplementation on blood pressure control and cardioprotective measures [31]. Moreover, omega-3 may also improve endothelial and vascular functioning and arterioles stiffness [32]. Few other studies conducted on underweight pregnant women showed no difference in systolic and diastolic blood pressure regarding LNS and IFA consumers, presumably due to the subjects being normotensive women [33,34].

Pregnancy outcomes were compared. It was observed that the number of preterm deliveries (<37 weeks) was 35% in Group 1, while all the women on LNS-PLW supplementation delivered after 37 weeks of gestation. The study reported that pre-eclampsia babies are usually preterm [35]. However, a study conducted by Wen et al., 2018, reported that supplementation with folic acid had no effect on babies’ gestational age and contradicts our finding [36]. The difference in our observations may be due to the additional intake of the LNS-PLW supplement along with IFA, whereas the study of Wen et al. assessed the effects of folic acid only [36].

The gestational age of our participants at enrollment (in both groups) was found to be below 34 weeks, meaning all were in the early-onset pre-eclampsia class. In the current study, it was observed that the numbers of live births are higher in the LNS-PLW consumers. Moreover, the number of intrauterine deaths (IUDs) is lower in the supplemented group. These findings are in line with another study that reported that supplementation in pre-eclampsia reduces IUD [37]. It was reported that multiple micronutrients (MMNs) supplementations reduced the risk of preterm birth, while LNS supplementation showed no positive effects on the term of pregnancy [13,27]. The number of NVDs was higher in Group 2 compared to Group 1, while 31% of the women in Group 1 had a caesarean section. It is documented that the rate of cesarean section increases with the severity of pre-eclampsia [38], and also micronutrient-deficient women have a greater risk of cesarean delivery [39]. Similarly, Kiattisak et al., 2018, reported that pre-eclamptic women had a higher occurrence of cesarean delivery [35,40]. In contrast to our findings, a study compared the pregnancy outcome of IFA and LNS consumer groups and found no significant difference in pregnancy and fetal outcomes regarding cesarean delivery in rural Bangladesh [41].

The strength of the current study is the participant’s compliance and cooperation in maintaining high-quality assurance during supplementation and data collection. Moreover, the low dropout rate was made possible by the main researcher herself collecting the data and following the participants.

This study has some limitations; firstly, the women in both groups were pre-eclamptic, and for comparison, normal pregnant women were not recruited. Secondly, due to time and resource constraints, we analyzed a small number of participants in only three antenatal units.

## 5. Conclusions

The prenatal use of LNS-PLW daily, along with IFA and regular follow-up, can improve pregnancies and maternal outcomes by increasing the live birth and term of the babies and decreasing maternal complications, like eclampsia and the number of deliveries by cesarean section.

## Figures and Tables

**Figure 1 medicina-58-01772-f001:**
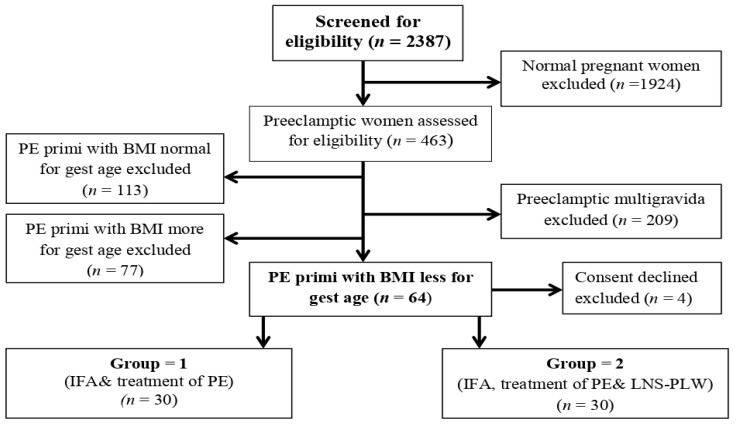
Flow chart showing the enrollment of the participants and intervention allocation. Abbreviations: PE: Preeclampsia, BMI: Basal metabolic index, IFA: Iron folic acid, LNS-PLW: Lipid based nutritional supplements for pregnant and lactating women.

**Table 1 medicina-58-01772-t001:** Chemical composition of (LNS-PLW).

Nutrient	Value	Nutrient	Value
Carbohydrate	35.4 g	Biotin (Vitamin-B7)	45 mcg
Protein	10.5 g	Folates (Vitamin-B9)	247 mcg
Fat (60%ω-3Fatty Acid,20%ω-6FA)	24 g	Cobalamine (Vit B12)	2.0 mcg
Retinol (Vitamin-A)	412 mcg	Iron	10 mg
Cholecalciferiol (Vitamin-D)	11.2 mcg	Calcium	400 mg
Phytomenadione (Vitamin-K)	20.2 mcg	Zinc	8.2 mg
Tocoferol acetate (Vit-E)	12 mg	Copper	1.0 mg
Ascorbate (Vitamin C)	45 mg	Selenium	15 mcg
Thiamine (Vitamin-B1)	0.75 mg	Iodine	75 mcg
Riboflavin (Vitamin-B2)	1.57 mg	Magnesium	112 mg
Niacin (Vitamin-B3)	9.75 mg	Phosphorus	337 mg
Pantothenic acid (Vit B5)	3.0 mg	Potassium	675 mg
Pyridoxine (Vitamin-B6)	1.35 mg	Manganese	0.9 mg
Total Energy = 400 kcal			

**Table 2 medicina-58-01772-t002:** Comparison of baseline parameters of the study groups.

Parameters	Group 1 *n* = 26 Mean ± SD	Group 2 *n* = 28 Mean ± SD	*p*-Value
Age (years)	22.1 ± 3.3	23.3 ± 3.6	0.17
Gest age at first antenatal visit retrospectively (weeks)	20 ± 3.5	20.5 ± 3.1	0.45
BMI Kg/m^2^ at 1st antenatal visit retrospectively	19.6 ± 1.3;	19.7 ± 0.9	0.84
Gestational age at enrollment in weeks	30.1 ± 2	29.8 ± 2.3	0.59
BMI Kg/m^2^ at enrollment	22.6 ± 1.5	22.5 ± 1.3	0.46
Systolic Blood Pressure mmHg	146.7 ± 8.9	144.7 ± 6.8	0.63
Diastolic Blood Pressure mmHg	95.7 ± 5.2	95.5 ± 4.4	0.89
Hb g/dL	10.5 ± 0.8	10.5 ± 0.9	0.92
platelets count/uL	202,166.67 ± 57,770.70	196,200.00 ± 53,832.05	0.68

**Table 3 medicina-58-01772-t003:** Effect of supplementation on maternal and pregnancy outcomes.

Parameters	Group 1 *n* = 26	Group 2 *n* = 28
Live birth	92%	93%
NVD C-section	18 (69%) 8 (31%)	22 (78%) 6 (22%)
Preterm babies	9 (35%)	-
IUD/stillbirth	2 (8%)	2 (7%)
LBW	16 (61.5%)	8 (28.5%)
Eclampsia	1 (4%)	-

Abbreviations: NVD: normal vaginal delivery; C-section: Caesarean section; IUD: intrauterine death; LBW: low birth weight.

**Table 4 medicina-58-01772-t004:** Clinical and hematological parameters of study groups at delivery.

Parameters	Group 1 *n* = 26 Mean ± SD	Group 2 *n* = 28 Mean ± SD	*p*-Value
Supplements consume (weeks)	6.8 ± 1.9	8.8 ± 1.9	0.00
Gestational age at delivery	36.9 ± 1.6	38.6 ± 0.8	0.00
Systolic Blood Pressure mmHg	135.7 ± 11.2	131.3 ± 4.6	0.07
Diastolic Blood Pressure mmHg	92.2 ± 5.2	89.6 ± 2.1	0.02
Hemoglobin g/dL	11.4 ± 0.5	12.2 ± 0.8	0.001
BMI(kg/m^2^) at delivery	27.17 ± 1.39	27.38 ± 0.94	0.570
Birth weight of the baby	2.37 ± 0.21	2.49 ± 0.08	0.003

## Data Availability

All data generated or analyzed during this work are included in this published article. Also, all data used to support the findings of this study are available from the corresponding author upon request.

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
