# Peer review of "Effect of Lipid-Based Multiple Micronutrients Supplementation in Underweight Primigravida Pre-Eclamptic Women on Maternal and Pregnancy Outcomes: Randomized Clinical Trial"

_medicina, 2022, doi:10.3390/medicina58121772_

Round 1

Reviewer 1 Report

In the present manuscript the authors evaluated the effect of a lipid-based multiple micronutrient supplementation in underweight preeclamptic women in term of maternal and, at least in part, perinatal outcome. The topic is interesting if we consider the incidence of preeclampsia and that, to date, the only cure is delivery. In this respect, any attempt to extend the gestational period to delivery and improve the outcome of pregnancy is welcome.

However, in my opinion, some information and revisions need to be included before the article can be suitable for publication in Medicina journal:

·        The Introduction section is quite poor and need to be enriched, especially in the first paragraph that it should also mention the two types of PE, early- and late- onset, which possess different etiopathogenesis and are actually considered two different pathologies. I suppose that the patients enrolled in this study are affected by early- onset PE, but for clarity the authors should make it explicit in the text.

·        In the first paragraph of the Introduction, I suggest also to treat the importance of an early diagnosis of preeclampsia and possible therapy. When the angiogenic imbalance is mentioned, the importance of the evaluation of multiple factors for an early and accurate diagnosis of PE is appropriate, i.e. sFLT-1, PLGF, sENDOGLIN, TGFb, EGFL7, there is a huge literature in this respect (see Hod T, et al. Molecular mechanisms of preeclampsia. Cold Spring Harb Perspect Med 2015;5:1–20; Levine RJ, et al. Circulating angiogenic factors and the risk of preeclampsia. N Engl J Med 2004;350:672–83; Massimiani M, et al. Increased circulating levels of Epidermal Growth Factor-like Domain 7 in pregnant women affected by preeclampsia. Transl Res. 2019;207:19-29). The evaluation of these factors allows also to discriminate between the two forms of PE and other pathologies (Perni U, et al. Angiogenic factors in superimposed preeclampsia: a longitudinal study of women with chronic hypertension during pregnancy. Hypertension 2012;59:740–6; Massimiani M, et al. Circulating EGFL7 distinguishes between IUGR and PE: an observational case-control study. Sci Rep. 2021;11(1):17919; Rolfo A, et al. Chronic kidney disease may be differentially diagnosed from preeclampsia by serum biomarkers. Kidney Int 2013;83:177–81).

·        Check for repetitions, e.g. “About 3-5% of first pregnancies are complicated by preeclampsia” and “It affects globally 3-5% of pregnant women”.

·        For clarity, the authors should state the definition of underweight and which cut-off of BMI was used in the study.

·        I suggest to rename the first paragraph of M&M “Patients” or “Study subjects” and include there the Inclusion and Exclusion Criteria, not considering them as two paragraphs. In this way, M&M would be formed by 3 paragraphs (patients, data collection and statistical analysis).

·        Move information given in lines 111-114 at the beginning of M&M.

·        Table 2 is never mentioned in the text. Please add a sentence as “Maternal findings at enrolment are reported in Table 2”.

·        Did you collect information regarding kidney health, e.g. protein:creatinine ratio? The athors should give this information both at enrolment and after treatment.

·        Data about the percentage of preterm babies, IUD are reported, but the neonatal birth weight is missing. It needs to be necessarily included to evaluate the effect of the diet on the perinatal outcome.

·        I suggest to include also the BMI at delivery to evaluate the maternal weight gain after the diet.

·        Discussion line 232: supplementation of what?

Author Response

Reply to Reviewers Reports

Reviewer 1 report

S no

comments

correction

1

Th Introduction section is quite poor and need to be enriched, especially in the first paragraph that it should also mention the two types of PE, early- and late- onset, which possess different etiopathogenesis and are actually considered two different pathologies. I suppose that the patients enrolled in this study are affected by early- onset PE, but for clarity the authors should make it explicit in the text.

Different types of Preeclampsia added to introduction. First paragraph of introduction line 9 to 14.

The entire participant of our study belongs to early-onset classed as discussed in fifth paragraph of Discussion.

2

In the first paragraph of the Introduction, I suggest also to treat the importance of an early diagnosis of preeclampsia and possible therapy. When the angiogenic imbalance is mentioned, the importance of the evaluation of multiple factors for an early and accurate diagnosis of PE is appropriate, i.e. sFLT-1, PLGF, sENDOGLIN, TGFb, EGFL7, there is a huge literature in this respect (see Hod T, et al. Molecular mechanisms of preeclampsia. Cold Spring Harb Perspect Med 2015;5:1–20; Levine RJ, et al. Circulating angiogenic factors and the risk of preeclampsia. N Engl J Med 2004;350:672–83; Massimiani M, et al. Increased circulating levels of Epidermal Growth Factor-like Domain 7 in pregnant women affected by preeclampsia. Transl Res. 2019;207:19-29). The evaluation of these factors allows also to discriminate between the two forms of PE and other pathologies (Perni U, et al. Angiogenic factors in superimposed preeclampsia: a longitudinal study of women with chronic hypertension during pregnancy. Hypertension 2012;59:740–6; Massimiani M, et al. Circulating EGFL7 distinguishes between IUGR and PE: an observational case-control study. Sci Rep. 2021;11(1):17919; Rolfo A, et al. Chronic kidney disease may be differentially diagnosed from preeclampsia by serum biomarkers. Kidney Int 2013;83:177–81).

Early screening and diagnosis and possible management is added to introduction in the second last paragraph.

3

Check for repetitions, e.g. “About 3-5% of first pregnancies are complicated by preeclampsia” and “It affects globally 3-5% of pregnant women”.

Repetition  removed

4

For clarity, the authors should state the definition of underweight and which cut-off of BMI was used in the study.

Pregnancy Weight Gain Calculator

https://www.calculator.net › fitness & health is used for BMI calculation and underweight for gestational age.

5

I suggest to rename the first paragraph of M&M “Patients” or “Study subjects” and include there the Inclusion and Exclusion Criteria, not considering them as two paragraphs. In this way, M&M would be formed by 3 paragraphs (patients, data collection and statistical analysis).

M&M reformed in 3 paragraph (patients, data collection and statistical analysis) according to suggestion.

6

 Move information given in lines 111-114 at the beginning of M&M.

Moved

7

Table 2 is never mentioned in the text. Please add a sentence as “Maternal findings at enrolment are reported in Table 2”.

Table-2 mentioned at the end of first paragraph of results

8

Did you collect information regarding kidney health, e.g. protein:creatinine ratio? The athors should give this information both at enrolment and after treatment.

Not  collected

9

Data about the percentage of preterm babies, IUD are reported, but the neonatal birth weight is missing. It needs to be necessarily included to evaluate the effect of the diet on the perinatal outcome.

The neonatal birth weight is added in Table-4

10

 I suggest to include also the BMI at delivery to evaluate the maternal weight gain after the diet.

BMI at delivery added in Table-4

11

  Discussion line 232: supplementation of what?

·       

Supplementation of LNS-PLW

Reviewer 2 Report

This study by Sher et al., shows that lipid based multiple micronutrients supplementation improves maternal and fetal outcomes in preeclamptic women. Please see my comments below.

Comments

1.      Please include the rationale for conducting the study in underweight preeclamptic women? Does this supplement improve outcomes in overweight or obese preeclamptic patients?

2.      Discuss why the supplement had an effect on diastolic but not systolic blood pressure? Please briefly discuss how this supplement effects maternal blood pressure (potential mechanism)

3.      What is the rational for including only primigravida preeclamptic women?

4.      Line 219- What do authors mean by “study conducted on under weight pregnant women showed no difference…………..” Please rewrite this to

5.      Line 114: Please clarify the name/weblink of computer randomizer

6.      Please fix the grammar and formatting throughout the text.

Author Response

Reviewer 2 report

S no

comments

correction

1

Th Extensive editing of English language and style required

Done

2

Please include the rationale for conducting the study in underweight preeclamptic women? Does this supplement improve outcomes in overweight or obese preeclamptic patients?

The rational for conducting the study in underweight preecmlamptic women has already been discussed (last paragraph of introduction). As overweight and obese preeclamptic women were not in inclusion criteria, so we do not know the effect of the supplements on such women. Future studies are required in order to investigate the effects of LNS in overweight and obese PE.

3

Discuss why the supplement had an effect on diastolic but not systolic blood pressure? Please briefly discuss how this supplement effects maternal blood pressure (potential mechanism)

Discussed in third paragraph of Discussion.

4

What is the rational for including only primigravida preeclamptic women?

Discussed in the end of last paragraph of introduction.

As we have already discussed the incidence of preeclampsia is 7.5% more in primigravida women therefore, such women were included in our study. MOREOVER in order to overcome other confounding factors in multigravidas we recruited primigravida preeclamptic women.

5

Line 219- What do authors mean by “study conducted on underweight pregnant women showed no difference…………..” Please rewrite this to

We compared our study with studies conducted on underweight pregnant women. And in those studies, there was no effect of supplementation on blood pressure as their subjects were normotensive.

6

 Line 114: Please clarify the name/weblink of computer randomizer

Random Number Generator - Calculator.net

https://www.calculator.net › math

7

  Please fix the grammar and formatting throughout the text.

Grammar and formatting fixed

Round 2

Reviewer 2 Report

No further comments